# miR-21-5p Under-Expression in Patients with Obstructive Sleep Apnea Modulates Intermittent Hypoxia with Re-Oxygenation-Induced-Cell Apoptosis and Cytotoxicity by Targeting Pro-Inflammatory TNF-α-TLR4 Signaling

**DOI:** 10.3390/ijms21030999

**Published:** 2020-02-03

**Authors:** Yung-Che Chen, Po-Yuan Hsu, Mao-Chang Su, Chien-Hung Chin, Chia-Wei Liou, Ting-Ya Wang, Yong-Yong Lin, Chiu Ping Lee, Meng-Chih Lin, Chang-Chun Hsiao

**Affiliations:** 1Division of Pulmonary and Critical Care Medicine, Department of Medicine, Kaohsiung Chang Gung Memorial Hospital and Chang Gung University College of Medicine, Kaohsiung 83301, Taiwan; yungchechen@yahoo.com.tw (Y.-C.C.); hsupowan@yahoo.com.tw (P.-Y.H.); maochangsu@yahoo.com.tw (M.-C.S.); chestman57112@yahoo.com.tw (C.-H.C.); filling.tw@yahoo.com.tw (T.-Y.W.); yonyonlin@yahoo.com.tw (Y.-Y.L.); choupeen@gmail.com (C.P.L.); 2Sleep Center, Kaohsiung Chang Gung Memorial Hospital and Chang Gung University College of Medicine, Kaohsiung 83301, Taiwan; 3Department of Medicine, Chang Gung University, Taoyuan 33302, Taiwan; 4Department of Respiratory Therapy, Chang Gung University of Science and Technology, Chia-Yi 61363, Taiwan; 5Department of Medicine, Chung Shan Medical University School of Medicine, Taichung 40201, Taiwan; 6Department of Neurology, Kaohsiung Chang Gung Memorial Hospital and Chang Gung University College of Medicine, Kaohsiung 83301, Taiwan; cwliou@ms22.hinet.net; 7Graduate Institute of Clinical Medical Sciences, College of Medicine, Chang Gung University, Taoyuan 33302, Taiwan

**Keywords:** obstructive sleep apnea, miR-21-5p, miR-23a-3p, apoptosis, intermittent hypoxia with re-oxygenation

## Abstract

The purpose of this study is to explore the anti-inflammatory role of microRNAs (miR)-21 and miR-23 targeting the TLR/TNF-α pathway in response to chronic intermittent hypoxia with re-oxygenation (IHR) injury in patients with obstructive sleep apnea (OSA). Gene expression levels of the miR-21/23a, and their predicted target genes were assessed in peripheral blood mononuclear cells from 40 treatment-naive severe OSA patients, and 20 matched subjects with primary snoring (PS). Human monocytic THP-1 cell lines were induced to undergo apoptosis under IHR exposures, and transfected with miR-21-5p mimic. Both miR-21-5p and miR-23-3p gene expressions were decreased in OSA patients as compared with that in PS subjects, while *TNF-α* gene expression was increased. Both miR-21-5p and miR-23-3p gene expressions were negatively correlated with apnea hypopnea index and oxygen desaturation index, while *TNF-α* gene expression positively correlated with apnea hypopnea index. In vitro IHR treatment resulted in decreased miR-21-5p and miR-23-3p expressions. Apoptosis, cytotoxicity, and gene expressions of their predicted target genes—including *TNF-α*, *ELF2*, *NFAT5*, *HIF-2α*, *IL6*, *IL6R*, *EDNRB*, and *TLR4*—were all increased in response to IHR, while all were reversed with miR-21-5p mimic transfection under IHR condition. The findings provide biological insight into mechanisms by which IHR-suppressed miRs protect cell apoptosis via inhibit inflammation, and indicate that over-expression of the miR-21-5p may be a new therapy for OSA.

## 1. Introduction

Obstructive sleep apnea (OSA) syndrome is characterized by episodes of partial or total upper airway obstruction during sleep with airflow interruption (apnea) or reduction (hypopnea), leading to recurrent events of transient reduction in oxyhemoglobin saturation (SaO_2_) and hypercapnia, followed by a transient awakening. Consequently, the patient generally presents fatigue, excessive daytime sleepiness, morning headache, attention or memory impairment, and habitual snoring history, and carries increased risks of hypertension, ischemic heart disease, diabetes mellitus, and cognitive dysfunction. Recent randomized controlled trials have not demonstrated significant protective effects of long-term continuous positive airway pressure treatment on cardiovascular events or mortality, indicating the need of developing pharmaceutical agents to improve outcomes in OSA patients [1,2,3].

It is known that the OSA pathogenesis is related to a multifactorial process with a diversity of mechanisms, including the sympathetic nervous system hyperactivity, intermittent hypoxia with re-oxygenation (IHR) induced excessive oxidative stress, vascular endothelial dysfunction, metabolic deregulation, and selective activation of the inflammatory responses. Toll-like receptor 4 (TLR4), a typical representative of pattern recognition receptors in innate immune responses, plays an important role in activation of inflammation in atherosclerosis. It has been recently demonstrated that IHR accelerated growth and vulnerability of atherosclerotic plaque, which probably acted by triggering the activation of TLR4/NF-κB signaling [4,5,6,7]. Acting as both up-stream and down-stream mediators of TLR signaling, tumor necrosis factor (TNF)-α has been shown to be upregulated in OSA patients, and this increase became more pronounced with the more severe grades of OSAS, indicating that TNF-α might be a promising circulating biomarker for the development of OSA [8]. In vitro experiments have shown that hypoxia enhance the response of endothelial and epithelial cells to oxidative stress through potentiating monocyte-derived dendritic cell and macrophage for release of TNF-α via MAP3K8 and ERK pathways [9,10,11,12]. Soluble TNF-α binds to TNFR1 receptor, resulting in the formation of complex I. When this complex dissociates, the death domains of TRADD and RIP1 are liberated and bind to FADD, which in turn recruits caspases 8 and 10, forming Complex II and culminating in apoptosis [13]. Previous studies have shown that cell viability of various end organs is decreased in the IHR environment, which is associated with the upregulation of apoptosis-related proteins.

MicroRNAs (miRNAs) are a group of endogenous non-coding single-strand RNAs with about 22 nucleotides in length, and inhibit their target gene expressions at the transcriptional or post-transcriptional level via complementary base pairing over 3’ un-translated region. However, the association between miRNAs and IHR-induced cell apoptosis and injury remains elusive. Human and animal studies have demonstrated that endothelial dysfunction is a precursor of atherosclerosis, and endothelial dysfunction observed in patients with OSA can occur through mechanisms including oxidative stress and activation of inflammation. Certain miRNAs, including miR-21 and miR-23a, are induced by TLR signaling or negatively regulate TLR signaling [14]. Specifically, miR-21 is involved in anti-inflammatory response via inhibiting TLR, NF-κB, IL-6, and TNF-α signaling pathway, while high concentrations of TNF-α inhibit miR-21 functional axis [15,16,17,18,19,20]. In contrast, miR-23a may be involved in pro-inflammatory response via enhancing TNF-α and IL-6 secretions, while TLR signaling can inhibit miR-23a expression [21,22,23]. In the current study, we made a comparison of miR-21/miR23a and their target genes between OSA patients and matched PS subjects, and used a cell culture model of IHR to demonstrate that TLR/TNF-α signaling-related microRNAs may be involved in the pathophysiological cell apoptosis and dysfunction under chronic IHR stimuli in OSA.

## 2. Results

A total of 20 PS subjects and 40 patients with treatment-naive OSA were enrolled and analyzed. Table 1 presents subjects’ demographic, PSG, and blood chemistry data. There were no significant differences between two groups in terms of age, sex, BMI, smoking history, co-morbidity, blood lipid profiles, and fasting blood sugar. There were significant differences in PSG parameters between two groups.

### 2.1. Decreased miR-21-5p/miR-23a-3p Gene Expressions and Increased TNF-α Gene Expression in OSA Patients

*TNF-α* gene expression was increased in OSA patients as compared with that in PS subjects (3.08 ± 4.25 vs. 1.32 ± 1.18 fold change, regression coefficient 1.913, 95%CI 0.037 to 3.456, adjusted *p* = 0.016, Figure 1a), and positively correlated with AHI (*r* = 0.388, *p* = 0.003, Figure 1b). MiR-21-5p gene expression was decreased in OSA patients versus PS subjects (0.08 ± 0.16 vs. 8.63 ± 18.24 fold change, regression coefficient −5.458, 95%CI −8.701 to −2.215, adjusted *p* = 0.002, Figure 1c), and negatively correlated with AHI (*r* = −0.478, p < 0.001, Figure 1d). MiR-23a-3p gene expression was decreased in OSA patients versus PS subjects (0.17 ± 0.26 vs. 7.17 ± 15.64, regression coefficient −5.018, 95%CI −7.653 to −2.384, adjusted p < 0.001, Figure 1e), negatively correlated with AHI(*r* = −0.446, *p* = 0.001, Figure 1f)/ODI(*r* = −0.267, *p* = 0.044, Figure 1h), and further decreased in those with morning headache (0.06 ± 0.05 vs.0.26 ± 0.34, regression coefficient −0.222, 95%CI-0.429 to −0.015, adjusted *p* = 0.037, Figure 1g). All statistics of the variables used in the adjusted analyses are listed in Table 1. 45% (18/40) of the OSA patients reported recurrent morning headache. Both miR-21-5p and miR-23a-3p gene expressions were negatively correlated with several predicted target gene expressions, including *TNF-α*, *TLR4*, *TLR6*, *NFAT5*, *ELF2*, *HIF-2α*, *EDNRB*, *SP1*, *PDCD4*, and *IRF1* (Table 2).

### 2.2. Effects of In Vitro IHR on the Two under-Expressed miRs and Their Predicted Target Gene Expressions in Human Monocytic THP-1 Cells

To determine whether IHR per se can affect the five miRs, their predicted target gene expressions, and cell apoptosis/viability, human monocytic THP-1 cell lines were exposed in vitro to either 7 cycles of IHR per day for 4 days or 4 days of continuous NOX condition. IHR resulted in increased early apoptosis marker on day 1 through day 3 (percentage of Annexin V + PI- cells, all *p*-values < 0.05, Figure 2a), increased late apoptosis marker on day 1 through day 3 (percentage of Annexin V+ PI+ double positive cells, all *p*-values < 0.05, Figure 2b), and decreased cell viability on day 2 through day 4 (WST-1 incorporation percentage, all *p*-values < 0.05, Figure 2c). 

To determine the target genes and molecular pathways related to miR-21-5p and miR-23-3p in apoptotic monocytes, the common targets and pathways of the two miRNAs downregulated by IHR were explored by the genes intersection option using the miRbase database. The results identified several miR-21-5p and miR-23-3p-regulated targets and pathways, most of which were involved in pro-inflammatory responses and thus selected for further evaluation. IHR treatment in vitro resulted in significantly decreased miR-21-5p gene expression on day 1 through day 4 (Figure 2d, all *p*-values < 0.05) and decreased miR-23-3p gene expression on day 1 and 4 (Figure 2e, both *p*-values < 0.05), as well as increased gene expressions of their predicted target genes, including *TNF-α* on day 1 through day 4 (Figure 2f, all *p*-values < 0.05), *TLR2* on day 2 through day 4 (Figure 2g, all *p*-values < 0.05), *TLR6* on day2 through day 4 (Figure 2h, all *p*-values < 0.05), and *HIF-2α* on day 2 through day 4 (Figure 2i, all *p*-values < 0.05).

### 2.3. MiR-21-5p Mimic Represses TNF-α, IL-6R, and TLR4 Gene Expressions in THP-1 Cells under NOX Condition in a Dose-Dependent Manner

Because only miR-21-5p was consistently downregulated both in OSA patients and in response to in vitro IHR exposures, it was selected for further functional studies. To investigate the protective effect of miR-21-5p on overt inflammation and cell injury, we first evaluated whether miR-21-5p mimic had inhibitory effects on several predicted target genes in THP-1 cells under NOX condition. miR-21-5p mimic at different concentrations (1, 5, 10, 25, and 50 nM, respectively) enhanced miR-21-5p gene expressions (Figure 3a, versus 0 nM, all *p*-values < 0.05) in a dose-dependent manner, while inhibited *TNF-α* (Figure 3b, versus 0 nM, all *p*-values < 0.05), *IL-6R* (Figure 3c, versus 0 nM, all *p*-values < 0.05), and *TLR4* (Figure 3d, versus 0 nM, all *p*-values < 0.05) gene expressions in a dose-dependent manner.

### 2.4. MiR-21-5p Mimic Transfection Suppressed IHR-Induced Cytotoxicity/Apoptosis and Inhibited IHR-Induced upregulations of Several Predicted Target Genes, Including TNF-α, HIF-2α, NFAT5, ELF2, IL6, IL6R, EDNRB, and TLR4

To explore the role of miR-21-5p in IHR-induced cytotoxicity and apoptosis, the THP-1 cells were transfected with 5 nM miR-21-5p mimic under IHR condition. Using the LDH assay and Annexin V/PI flowcytometry analysis, we investigated whether miR-21-5p mimic could protect THP-1 cells from cell death and apoptosis under IHR environment. After 2 days of IHR exposures, the percentage of cytotoxicity in THP-1 cells were increased as compared with that in NOX condition (37.18 ± 1 versus 8.17 ± 0.13%, *p* = 0.021, Figure 3e), and decreased with miR-21-5p mimic transfection under IHR condition as compared with that in IHR alone condition (19.25 ± 1.86%, *p* = 0.021, Figure 3e). The percentage of early apoptosis marker was increased as compared with that in NOX condition (69.66 ± 4.6 vs. 1.76 ± 0.25%, *p* < 0.05), and decreased with miR-21-5p mimic transfection under IHR condition (44.33 ± 15.65%, *p* < 0.05, Figure 3f) as compared with that in IHR alone condition. Several predicted target genes, including *TNF-α* (Figure 3g), *HIF-2α* (Figure 3h), *NFAT5* (Figure 3i), *ELF2* (Figure 3j), *IL6* (Figure 3k), *IL6R* (Figure 3l), *EDNRB* (Figure 3m), and *TLR4* (Figure 3n), were significantly upregulated by IHR (all *p*-values < 0.05), and the increased expressions of these genes were suppressed by miR-21-5p mimic transfection (all *p*-values < 0.05). 

## 3. Discussion

The present results demonstrated decreased levels of miR-21-5p and miR-23-3p and increased levels of *TNF- α* both in OSA patients and in IHR-induced apoptotic monocytes. In vitro studies revealed that miR-21-5p protected monocyte under an IHR environment by inhibiting inflammation associated gene expressions to reduce cytotoxicity and apoptosis. The findings further pointed to rescuing the downregulated miR-21 levels as a potential strategy to protect monocytes from IHR-induced inflammation and apoptosis and gave reasons to suspect that TNF- α, IL-6R, and TLR4 mediated some of the therapeutic benefits of miR-21-5p mimic.

Previous studies have shown that miR-21 can inhibit cardiac myocyte/hepatic stellate cell apoptosis through PTEN/PI3K/Akt pathway, protect against neuronal cell death through FasL signaling, attenuate angiogenesis through SMAD7 signaling, and suppress secretion of inflammatory cytokines and chemokine receptor type 7 under hypoxia conditions, whereas it may contribute to pulmonary hypertension through inhibiting DDAH1 and RhoB under hyoxia [24,25,26,27,28]. Recently, miR-21 has been demonstrated to protect IHR or ischemia-reperfusion injury to islet cell, hepatocyte, nucleus pulposus cell, cardiac cells, and kidney epithelial cell via inhibiting PTEN/PI3K/AKT signaling pathway [29,30,31,32]. Furthermore, miR-21 could prevent human neural stem cell injury induced by oxygen and glucose deprivation through PDCD4/caspase-3 pathway [33,34]. In contrast, miR-21 over-expression was associated with right ventricle dysfunction in patients with hypoxia-induced pulmonary hypertension, and promoted neuronal damage in vitro [26,35]. In the present study, we found decreased miR-21-5p gene expressions both in severe OSA patients and in response to IHR stimuli. Moreover, miR-21-5p might protect monocyte from IHR-induced cell apoptosis and cytotoxicity by inhibiting TLR4/TNF-α/IL-6R-mediated inflammation. In line with our findings, previous studies have shown that miR-21 has anti-inflammatory and anti-apoptosis effects by inhibiting NF-κB-TNF-α-TLR and PDCD4-caspase-3 pathways, respectively, while TNF-α can suppress miR-21 functional axis [16,17,18,20,25,29,36]. The results open the possibility of using miR-21-5p mimic to overcome systemic inflammation and improve outcomes in OSA patients. Further mechanic studies are required to clarify its direct targets under IHR condition.

MiR-23a can protect retinal pigment epithelial cells against oxidative injury through regulation of Fas, alleviate hypoxia-induced neuronal apoptosis through suppressing Apaf-1, and inhibit vascular permeability of endothelial cells through regulation of JAM-C/ZO-2, whereas it may contribute to myocardial dysfunction due to ischemia-reperfusion injury through regulation of glutamine metabolism, facilitate cell proliferation and migration by targeting BMPR2/Smad1 signaling in hypoxia-induced human pulmonary artery smooth muscles, and potentiate hypoxia-induced injury of cardimyocytes [37,38,39,40,41]. On the other hand, miR-23a has been shown to be downregulated in mice under short-term IHR exposure, and inhibition of miR-23a in stressed cells represents a general mechanism for inducing apoptosis through targeting pro-apoptotic genes [42,43]. In the present study, we found decreased miR-23a-3p gene expressions in severe OSA patients and partly downregulation in response to IHR stimuli, and it was further decreased in those with morning headache. There is a strong association between OSA and ‘sleep apnea headache’, which is described as a recurrent morning headache, with resolution within 72 h of effective treatment for sleep apnea. Morning headache has been reported in OSA patients with variable rates ranging from 15 to 60%, but mixed results are present in the literature regarding headache improvement by means of continuous positive airway pressure treatment [44]. The cause of sleep apnea headache remains to be elucidated, because oxygen desaturation alone cannot explain the pathophysiology of sleep apnea headache [45]. Our results suggest that miR-23a-3p under-expression may be a novel mechanism that could explain why morning headache develops in only around one third of OSA patients. However, underlying mechanisms and the cause and effect relationship need further clarification.

Several limitations in the current study should be acknowledged. First, the negative correlations between the two miRs and their target genes are relatively weak. Because many clinical factors may potentially affect these gene expression levels, we used multivariate linear regression analysis models to minimize the effects of the confounding factors. Mir-21-5p, miR-23a-3p, and *TNF-α* remained statistically different between the OSA and PS groups, although some variables, such as BMI and smoking history, also showed independent associations with these gene expressions. Second, the IHR protocol could not mimic closely the clinical scenario in OSA patients. Because OSA is a disease characterized by chronic IHR, intermittent hypercapnia, sleep fragmentation, and sympathetic hyperactivity, it is difficult to mimic closely clinical scenario of OSA either using in vitro experiments or in vivo animal models. Previous studies have shown that intermittent hypoxia could be achieved by repeated 25 min exposure to 21% O_2_ + 5% CO_2_ followed by 25 min exposure to 0% O_2_ + 5% CO_2_ [46,47], We adopted this protocol with some modifications to increase IHR frequencies, which may theoretically mimic more closely IHR condition in OSA patients. Third, the differences of miR-23-3p and some predicted target genes between NOX and IHR conditions in vitro were not consistently separated from day 1 through day 4 experiments. However, increased early and late apoptosis markers were found in the first 3 days of IHR stimuli, while decreased cell viability was found in the later 3 days. We speculate that multiple signaling pathways may be involved in the regulations of IHR-mediated cell apoptosis and death in an undetermined sequence. Finally, further luciferase reporter analysis is required to confirm which genes are the direct targets of miR-21-5p.

In conclusions, we found miR-21-5p and miR-23a-3p were both downregulated in treatment-naive severe OSA patients, while *TNF-α* was upregulated, with all these three gene expressions being correlated well with disease severity. The results of the in vitro experiments revealed a new function of miR-21, suggesting that miR-21-5p-TLR4-TNF-α-IL6R signaling may act as a modulator to protect cell from apoptosis and cytotoxicity through inhibiting the activity of pro-inflammatory genes under IHR stimuli. The findings point out a promising therapeutic strategy for adverse consequences of OSA.

## 4. Materials and Methods

### 4.1. Subjects

The study was approved by the Institutional Review Board of Chang Gung Memorial Hospital, Taiwan (certificate number: 104-9756B, approval date: 23/02/2016. Written informed consent was obtained from all subjects participating in the study, who were aged 20 years or older and present with loud snoring. The study participants were recruited from the sleep center and pulmonary clinics of Kaohsiung Chang Gung Memorial Hospital during the period from August 2016 through July 2019. Eighty-eight subjects with snoring who underwent full-night PSG exam were screened initially. Exclusion criteria included ongoing infections, any known autoimmune disease, recent use of immunosuppressive agent in the past half year, narcolepsy, morbid obesity (body mass index, BMI >34 kg/m^2^), too old age (>65 years old), and too lean body weight (BMI < 21 kg/ m^2^). A total of 60 subjects were enrolled for final analysis, including 20 subjects with primary snoring (PS, apnea hypopnea index, AHI, <5 events/hour), and 40 treatment-naive patients with severe OSA (AHI ≥ 30 events/hour), Co-morbidities or adverse consequences of OSA, including hypertension (defined as baseline blood pressure > 140/90 mmHg), cardiac disease, diabetes mellitus, stroke, and neurologic deficits, were recorded.

### 4.2. Overnight Polysomnography (PSG) Examination

All the study participants underwent overnight PSG examination at the sleep center of Kaohsiung Chang Gung Memorial Hospital. A Sandman SD32 + TM Digital Amplifier (Embla, Nellcor Puritan Bennett, Boulder, CO, USA.) was used to conduct PSG measurements. Sleep stage scoring was performed by trained technicians according to standard criteria [48]. Nocturnal hypoxemia was evaluated in terms of mean SaO_2_, minimum SaO_2_, and the number of dips >4% of basal SaO_2_%//h (oxygen desaturation index, ODI). Snoring sampling was performed at 10 to 50 Hz, and all sounds of more than 50% above baseline amplitude lasting for 0.5–5 s were recorded by Piezo crystal snore sensor (SleepSense, Scientific Laboratory Products, Elgin, IL, USA). The Epworth Sleepiness Scale (ESS) recorded at the PSG exam was used to measure sleep propensity in every study subject [49].

### 4.3. Isolation of Leukocyte RNA

Peripheral blood mononuclear cells (PBMCs) were isolated from heparinized blood of all study subjects using a two-layer Ficoll-Histopaque density gradient centrifugation (Histopaque 1.077 and 1.119; Sigma Diagnostics, St. Louis, MO, USA) method. An RNeasy^®^Plus Mini Kit (Qiagen, Hilden, Germany) was used for isolation of high quality total RNA, and treated with Dnase according to the manufacture protocol. 

### 4.4. Analysis of miRNA

cDNA was generated from 2 µL of purified total RNA using the TaqMan Advanced miRNA cDNA Synthesis kit (Thermo Fisher Scientific, Waltham, MA, USA). Additionally, 1 pM of the synthetic *C. Elegans* oligo, cel-miR-39 (Sequence: UCACCGGGUGUAAAUCAGCUUG), was added to the isolated total RNA. This sequence does not exist in humans and was used as an exogenous control. All qPCR reactions were normalized to their corresponding cel-miR-39 *C*_t_ values. Quantitative RT-PCR was performed for each sample using 2.5 µL of diluted cDNA, TaqMan Advanced miRNA Assays (cel-miR-39-3p: 478293_mir; hsa-miR-21-5p: 477975_mir; hsa-miR-23a-3p: 478532_mir; Thermo Fisher Scientific), and Applied Biosystems™ TaqMan™ Fast Advanced Master Mix (Thermo Fisher Scientific) under fast cycling conditions. All TaqMan assays quantitative RT-PCR was carried out using the ABI 7500fast Real-Time PCR System (Applied Biosystems, Waltham, MA, USA). Real-time PCR cycling conditions consisted of 95 °C for 20 s, followed by 40 cycles of 95 °C for 3 s and 60 °C for 30 s. All miRNA fold changes were determined by the 2^−ΔΔ*C*T^ method.

### 4.5. Determination of Target Gene mRNA Expressions of Isolated PBMCs Using Quantitative Reverse-Transcriptase Polymerase Chain Reaction (RT-PCR)

To determine the expressions of the predicted target genes, the gene expressions of *TNF-α*, *TLR2*, *TLR4*, *TLR6*, *IL6*, *IL6R*, *NFAT5*, *ELF2*, *HIF-2α*, *EDNRB*, *SP1*, *PDCD4*, and *IRF1* were analyzed using quantitative RT-PCR in a 96-well format. The house keeping gene *GAPDH* was chosen as an endogenous control to normalize the expression data for each gene. All PCR primers (random hexamers) were designed and purchased from Roche according to the company’s protocols (www.roche-applied-science.com), and their sequences are given in Table 1. RNA samples were treated with DNAfree to remove contaminating genomic DNA. A total of 300 ng RNA was used for synthesis of first strand cDNA with QuantiTectReverse Transcription Kit (QIAGEN, Hilden, Germany). A total of 5 μL of the reverse transcription reaction was added to 5 μL of master mix (QIAGEN, SYBR Green PCR kit, Hilden, Germany; Roche, Grenzach-Wyhlen, Germany). The PCR reactions with 45 cycles of amplification were run in a Roche LightCycle 480 machine. Single real time PCR experiment was carried out on each sample for each target gene, because the Roch Light CyclerQuantiFast R system has shown high reproducibility. Relative expression levels were calculated using the 2^−ΔΔ*C*T^ method with the median value for the control group as the calibrator.

### 4.6. Transfection of miRNA-21-5p Mimic

miR-21-5p mimic (final concentration, 5 nM) was synthesized by GenePharma, and was incubated in THP-1 cells with Lipofectamine 2000 (Invitrogen, Carlsbad, CA, USA) for 6 h to over-express the gene expression level of miR-21-5p using the HiPerFect transfection reagent (QIAGEN, Hilden, Germany). The efficiency of the transfection was detected by quantitative RT-PCR.

### 4.7. In Vitro IHR Stimuli

THP-1 cell were exposed to IHR or normoxic (NOX) condition in a custom-designed, incubation chambers which were attached to an external O_2_–CO_2_ hand-driven controller. IHR protocol consists of a 17-min hypoxic period (0% O_2_ and 5% CO_2_) and 13 min of re-oxygenation period (21% O_2_ and 5% CO_2_) per cycle, 2 cycles/hour, 8 cycles/day for 1–4 days. We adopted this IHR protocol because previous studies have shown that a 30–40% decreases in blood SaO_2_ could be achieved in the conditioned media by 25-min of continuous exposure of cells to 0% O_2_ and 5% CO_2_ [46,47]. We restricted the hypoxic period to 17 min to increase IHR cycles and mimic more closely the clinical scenario in OSA patients.

### 4.8. Measurement of Cell Apoptosis by Flow Cytometry Analysis

Following treatment, THP-1 cell apoptosis rates were evaluated by flow cytometry using an Annexin V/Propidium iodide (PI) apoptosis detection kit (BD Biosciences, Franklin Lakes, NJ, USA). Cells were washed twice with PBS, re-suspended in binding buffer and incubated with 5 μL FITC-Annexin V and 5 μL PI for 15 min at room temperature. Staining cells were analyzed using the FACScan flow cytometry system (Becton Dickinson, San Diego, CA, USA).

### 4.9. Measurement of Cell Viability (Mitochondrial Activity) by WST-1

WST-1 reagent (Roche, Mannheim, Germany) diluted 1:10 in growth medium was added into THP-1 cells grown in a 96-well plate (10^4^ cells/well) for the last 1 h according to the manufacturer’s instructions. The amount of viable cells was determined via optical density measurement using a microplate reader at 450 nm, with 600 nm as a reference wavelength.

### 4.10. Measurement of Cytotoxicity (Cell Membrane Int.actness) by LDH Assay

Lactate dehydrogenase (LDH) assay Pierce™ LDH Cytotoxicity Assay Kit (Thermo Scientific) was performed according to manufacturer’s instructions. Briefly, after 2 days of exposure to IHR, THP-1 cells were incubated with 100 µL of (LDH reaction buffer  +  substrate) for 30 min at room temperature, followed by 50  µL of stop solution. The absorbance was read at 490 nm (iMarkTM Microplate Absorbance Reader, Bio-Rad, Hercules, CA, USA). Cytotoxicity percentage was calculated according the following formula: (treated sample LDH activity-spontaneous LDH release control activity)/(maximum LDH release control activity-spontaneous LDH release control activity) × 100%. 

### 4.11. Statistical Analysis

Continuous values were presented as the mean ± standard deviation (SD). Student’s *t*-test or Mann–Whitney U test was used for comparing mean values or distributions of two clinical or experimental groups where appropriate. Kruskal–Wallis test followed by post-hoc analysis was used to compare distributions of more than two experimental groups. Chi-square tests were used to assess the differences of category values between different groups. Multivariate linear regression model was used to adjust for potential confounding factors, including age, gender, BMI, co-morbidities (hypertension, diabetes mellitus, stroke, cardiac disease, and chronic kidney disease), smoking and alcoholism history, and to obtain regression coefficient, 95% confidence interval (CI), and adjusted *p*-values. Spearman’s correlation test was used to assess the correlation between two continuous variables. All tests were two tailed and the null hypothesis was rejected at *p* < 0.05. A statistical software package (SPSS, version 15.0, Chicago, IL, USA) was used for all analyses.

## Figures and Tables

**Figure 1 ijms-21-00999-f001:**
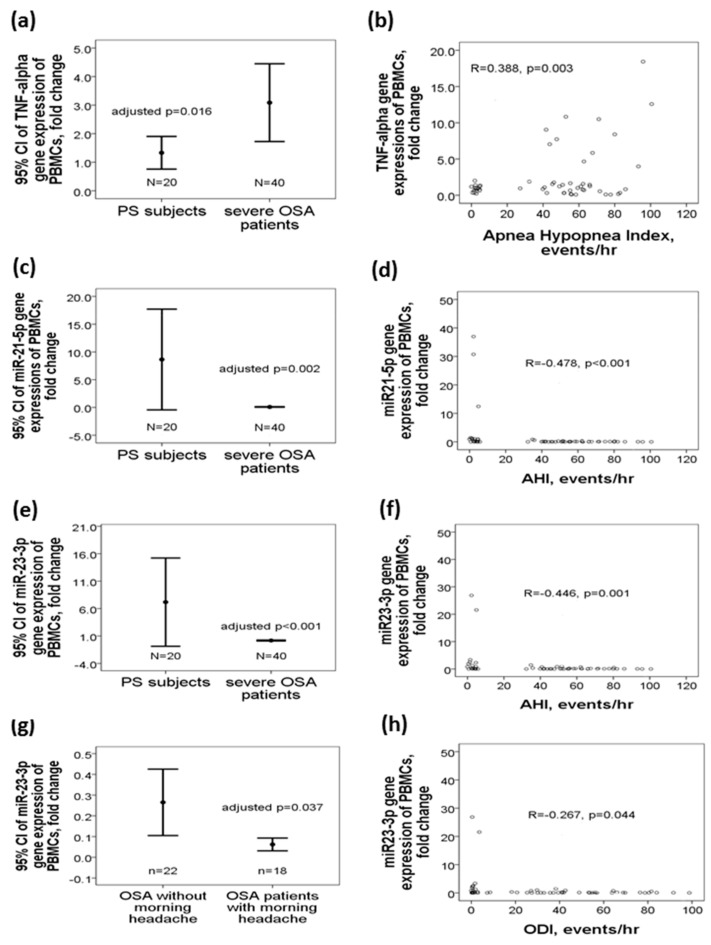
Upregulations of miR-21-5p/miR-23a-3p and downregulation of *TNF-α* in patients with treatment-naive obstructive sleep apnea. *TNF-α* gene expression was (**a**) increased in OSA patients, and (**b**) positively correlated with apnea hyponea index. miR-21-5p gene expression was (**c**) decreased in OSA patients, and (**d**) negatively correlated with apnea hypopnea index. miR-23a-3p gene expression was (**e**) decreased in OSA patients, negatively correlated with (**f**) apnea hypopnea index/(**h**) oxygen desaturation index, and (**g**) further decreased in those with morning headache.

**Figure 2 ijms-21-00999-f002:**
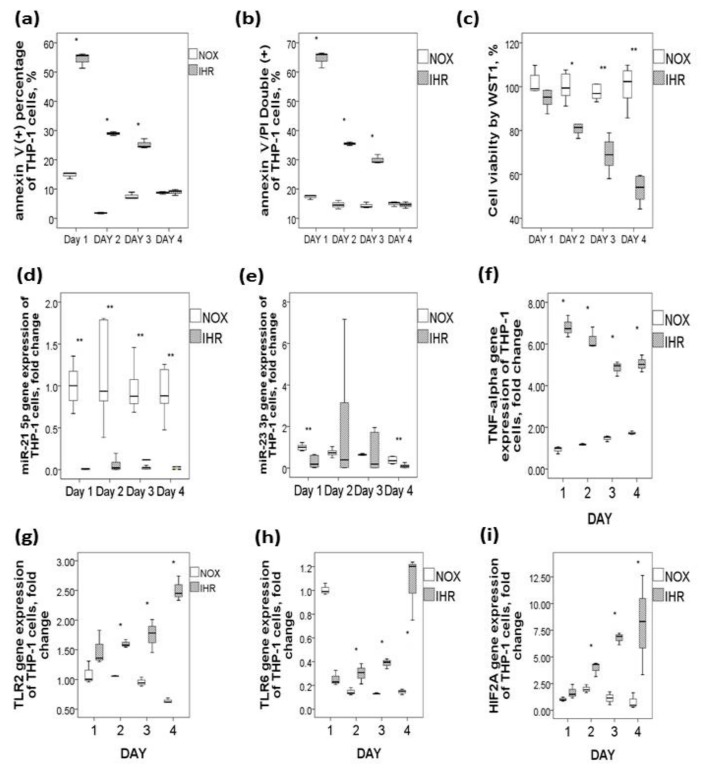
Downregulations of miR-21-5p/miR-23a-3p and upregulations of their predicted genes along with increased apoptosis in response to intermittent hypoxia with re-oxygenation (IHR) stimuli versus normoxic (NOX) condition in THP-1 cells. IHR in vitro for 1–4 days resulted in (**a**) increased early apoptosis marker, (**b**) increased late apoptosis marker, (**c**) decreased cell viability, (**d**) decreased miR-21-5p expression, (**e**) decreased miR-23a-3p expression, (**f**) increased *TNF-α*, (**g**) increased *TLR2*, (**h**) increased *TLR6*, and (**i**) increased *HIF2-α* gene expressions. * *p* < 0.05, compared between IHR and NOX condition by Mann–Whitney test; ** *p* < 0.01, compared between IHR and NOX condition by Mann-Whitney test.

**Figure 3 ijms-21-00999-f003:**
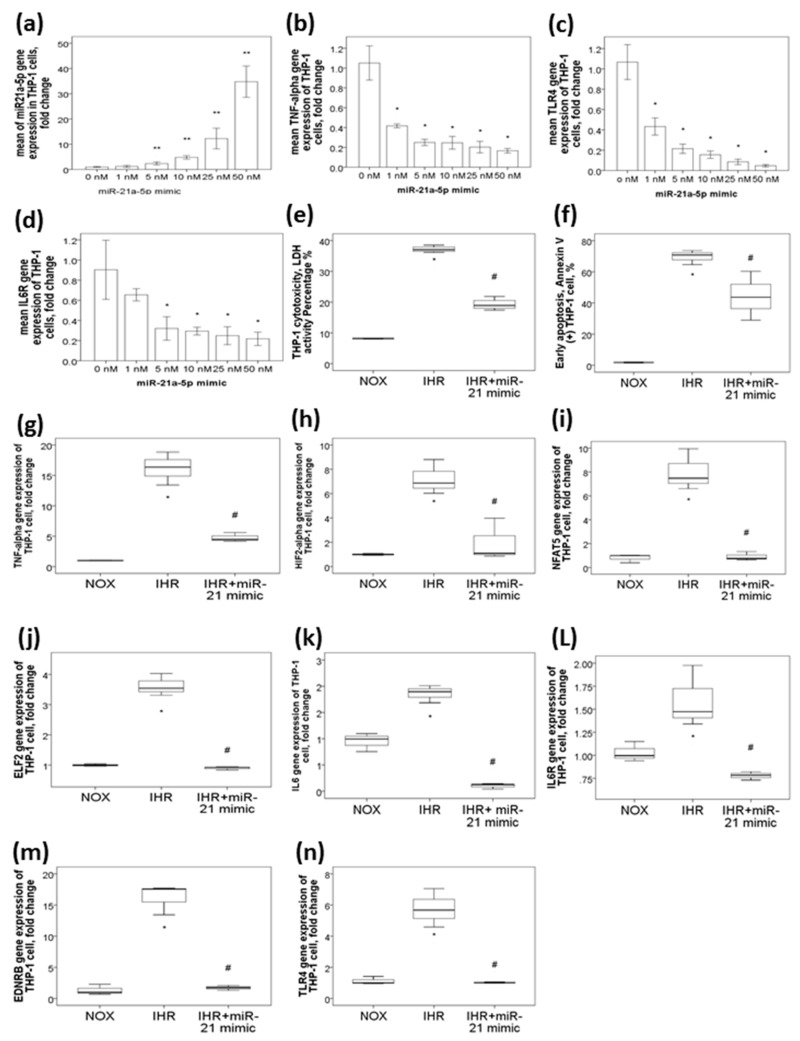
miR-21-5p mimic transfection suppressed IHR-induced cytotoxicity/apoptosis, and inhibit IHR-induced upregulations of several predicted target genes, including *TNF-α*, *HIF-2α*, *NFAT5*, *ELF2*, *IL6*, *IL6R*, *EDNRB*, and *TLR4*. miR-21-5p transfection (**a**) enhanced miR-21-5p expression and suppressed (**b**) *TNF-α*, (**c**) *TLR4*, and (**d**) *IL6R* gene expressions under normoxic (NOX) condition in a dose-dependent manner. Intermittent hypoxia with re-oxygenation (IHR) augment (**e**) cytotoxicity, (**f**) early apoptosis, and increased gene expressions of the (**g**) *TNF-α*, (**h**) *HIF2-α*, (**i**) *NFAT5*, (**j**) *ELF2*, (**k**) *IL6*, (**l**) *IL6R*, (**m**) *EDNRB*, and (**n**) *TLR4* genes, while all of these changes were reversed with miR-21-5p transfection under IHR condition. * *p* < 0.05, compared with 0 nM miR-21-5p mimic or NOX condition. ** *p* < 0.01, compared between IHR and NOX condition by Mann-Whitney test. # *p* < 0.05, compared with IHR alone condition.

**Table 1 ijms-21-00999-t001:** Demographic, biochemistry, and sleep data of all the 60 study participants

	PS Subjects (*n* = 20)	Severe OSA Patients (*n* = 40)	*p*-Value
Age, years	44.6 ± 14.3	47.6 ± 11	0.392
Male Sex, *n* (%)	15 (75)	33 (82.5)	0.432
Body mass index, kg/m^2^	25 ± 3.1	25.9 ± 2.7	0.262
AHI, events/hour	2.8 ± 1.6	60.7. ± 11.3	<0.001
ODI, events/hour	1.1 ± 0.9	48.9 ± 22.7	<0.001
Mean SaO_2_, %	96.2 ± 1.3	93.3 ± 2.8	<0.001
Minimum SaO_2_, %	89.2 ± 3.5	70 ± 12.6	<0.001
Snoring index, counts/hour	114 ± 141	331 ± 145	<0.001
Epworth Sleepiness Scale	8.5 ± 4.5	11.6 ± 5.2	0.034
Excessive daytime sleepiness, *n* (%)	5 (25)	21 (53.8)	0.048
Smoking history, *n* (%)	4 (20)	15 (37.5)	0.17
Cholesterol, mg/dL	189.5 ± 65	192.6 ± 32.3	0.822
Triglycerides, mg/dL	155.3 ± 224.9	149.9 ± 77.8	0.893
Hypertension, *n* (%)	5 (25)	16 (40)	0.305
Diabetes mellitus, *n* (%)	2 (10)	4 (10)	1
Heart disease, *n* (%)	3 (15)	2 (5)	0.186
Stroke, *n* (%)	1 (5)	0	0.143
Chronic kidney disease, *n* (%)	1 (5)	1 (2.5)	0.584

PS = primary snoring; AHI = apnea hypopnea index; ODI = oxygen desaturation index; SaO_2_ = arterial oxyhemoglobin saturation.

**Table 2 ijms-21-00999-t002:** Both miR-21-5p and miR-23a-3p gene expressions are negatively correlated with several predicted target gene expressions.

microRNA/Predicted Target Gene	TNF-α	TLR4	TLR6	NFAT5	ELF2	HIF-2α	EDNRB	SP1	PDCD4	IRF1	Correlation Coefficient/*p* value
miR-21	−0.431	−0.386	−0.436	−0.473	−0.569	−0.429	−0.545	−0.604	−0.653	−0.359	*R*
	<0.001	0.003	<0.001	<0.001	<0.001	<0.001	<0.001	<0.001	<0.001	0.005	*p*
miR-23a	−0.356	−0.346	−0.374	−0.377	−0.493	−0.366	−0.505	−0.551	−0.612	−0.276	*R*
	0.001	0.008	0.001	0.004	<0.001	0.001	<0.001	<0.001	<0.001	0.038	*p*

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
