# Peer review of "miR-21-5p Under-Expression in Patients with Obstructive Sleep Apnea Modulates Intermittent Hypoxia with Re-Oxygenation-Induced-Cell Apoptosis and Cytotoxicity by Targeting Pro-Inflammatory TNF-α-TLR4 Signaling"

_ijms, 2020, doi:10.3390/ijms21030999_

Round 1

Reviewer 1 Report

This is an important manuscript regarding the role of intermittent hypoxia and its effect on cells. There are a number of concerns that need to be addressed

Major concerns

In Table 1, only 3/40 OSA patients were male and it is mentioned that these accounted for 82.5% of severe OSA patients. Is this a typo and actual numbers are not listed. Similarly the p value for smoking between the two groups is reported as insignificant even though nearly 40% of patients in severe OSA group were smokers. Smoking can for affect hypoxia responses and it is difficult to interpret these results in smokers. Ideally these experiments should be done in non-smokers with and without sleep apnea. The correlation coefficient is weak for both miR21 and miR23 when looking at gene expression for TNF, TLR-4/6, NFAT5, HIF2a, IRF1 and also when looking at AHI. This should be mentioned. Additionally, is there any data showing relationship between oxygen desaturation indices (4%) and these microRNA expressions? In Figure 2, the graphs showing early and late apoptosis are difficult to interpret as the IHR and NOX values are close (especailly when the CIs are included. Similarly, the graphs showing gene expressions of miR-23-3p , TNFa, TLR6, HIF2a do not show the clear separation of values between IHR and NOX samples. One of the biggest problems with this study is the IHR protocol that includes 17 minutes of anoxia with 0% of O2 and 5% CO2 (this is not hypoxia) and 13 minutes of hyperoxia with 21% of O2 (this is not normoxia. No cell in the human body is exposed to 21% O2 - this represents oxygen concentrations in the hyperoxic range for the cell. Additionally 2 cycles/hr and 8 cycles/day (without specific timing for this exposures) is not a pattern that is seen in the clinical scenario. There should be explanation as to why such a protocol was chosen. for figures Figure 3e-n, how much of the miR-21-5p mimic was used? It shows in the figures 3a-d that up to 50nM of miR-21a-5p mimic was required.  While the transfection of miR-21-5p downregulated inflammatory genes, additional way to understand the effect of miR-21-5p is to transfect the inhibitor of miR-21-5p to see if that upregulates inflammatory gene expression. There are two forms of miR-21p that include miR21-5p and miR21-3p. Was miR21-3p also tested? The role of miR23a is unclear as described in introduction. Therefore the description of miR23a should be reduced. The discussion also delves into myriad pathways that miR21 affects that are not being measured in this study. Therefore authors should confined discussion to effects on NF-kB/TLR pathways that have studied in this manuscript.

MINOR COMMENTS

Abstract: This should be broken down into Rationale, methods, results and conclusions. If this is not required by the journal, these should be separated into different paragraphs.

Introduction: The language need to be addressed. few examples are

Para 1, Randomized controlled trials have demonstrated protective effects from... Para 2, and this increase is more pronounced with higher severiies of OSAS indicating that TNFa may be a ... Para 2, line on "In vitro experiments.. " does not make sense

Author Response

Response to Reviewer 1

This is an important manuscript regarding the role of intermittent hypoxia and its effect on cells. There are a number of concerns that need to be addressed

Major concerns

In Table 1, only 3/40 OSA patients were male and it is mentioned that these accounted for 82.5% of severe OSA patients. Is this a typo and actual numbers are not listed. Similarly the p value for smoking between the two groups is reported as insignificant even though nearly 40% of patients in severe OSA group were smokers. Smoking can for affect hypoxia responses and it is difficult to interpret these results in smokers. Ideally these experiments should be done in non-smokers with and without sleep apnea.

Ans.: Thanks for your comments. Actually, 33/40 OSA patients were male. We correct this typo error. To adjust the effects of several potential confounding factors, we made a linear regression analysis for the comparisons of the miRs and their target genes between OSA and PS groups. The statistical results, including regression coefficient, 95% CI, and adjusted p values, are listed in Table 1. miR-21, miR23a, and TNF-alpha still show significant differences between the 2 groups after adjusted for these factors, although their expression levels were also affected by some other clinical factors.

The correlation coefficient is weak for both miR21 and miR23 when looking at gene expression for TNF, TLR-4/6, NFAT5, HIF2a, IRF1 and also when looking at AHI. This should be mentioned. Additionally, is there any data showing relationship between oxygen desaturation indices (4%) and these microRNA expressions?

Ans.: As suggest, we mention the weak correlations in the discussion section. MiR-23a-3p gene expression was negatively correlated with AHI(r=-0.446, p=0.001, Figure 1(f)) /ODI(r=-0.267, p=0.044, Figure 1(h)), as described in the result 2.1 section.

In Figure 2, the graphs showing early and late apoptosis are difficult to interpret as the IHR and NOX values are close (especailly when the CIs are included. Similarly, the graphs showing gene expressions of miR-23-3p , TNFa, TLR6, HIF2a do not show the clear separation of values between IHR and NOX samples.

Ans.: Early and late apoptosis markers both showed significantly increased with IHR stimuli on day 1 through day 3 but not on day 4, whileTLR6, and HIF2a showed significant differences on day 2 through day 4 but not on day 1. However, TNF-alpha showed significantly increased with IHR on day 1 through day 4. The time lag may reflect different grades of importance of the predicted target genes in regulating cell apoptosis. miR-23-3a showed significant depressed only on day 1 and day 4 of IHR exposure, so it was not selected for further transfection experiments.

One of the biggest problems with this study is the IHR protocol that includes 17 minutes of anoxia with 0% of O2 and 5% CO2 (this is not hypoxia) and 13 minutes of hyperoxia with 21% of O2 (this is not normoxia. No cell in the human body is exposed to 21% O2 - this represents oxygen concentrations in the hyperoxic range for the cell. Additionally 2 cycles/hr and 8 cycles/day (without specific timing for this exposures) is not a pattern that is seen in the clinical scenario. There should be explanation as to why such a protocol was chosen.

Ans.: OSA is a disease characterized by chronic IHR, sleep fragmentation, and sympathetic hyperactivity. Thus, it is difficult to mimic clinical scenario of OSA either using in vitro experiment or in vivo animal models. We adopted this IHR protocol because previous studies have shown that a 30–40 % decreases in blood SaO2 could be achieved in the conditioned media by 25-min of continuous exposure of cells to 0%O2 and 5%CO224,25. We restricted the hypoxic period to 17 min to increase IHR frequencies and mimic more closely the clinical scenario in OSA patients. We add this statement along with two references in the method 4.7 section, and acknowledged this limitation in the discussion section.

for figures Figure 3e-n, how much of the miR-21-5p mimic was used? It shows in the figures 3a-d that up to 50nM of miR-21a-5p mimic was required.  While the transfection of miR-21-5p downregulated inflammatory genes, additional way to understand the effect of miR-21-5p is to transfect the inhibitor of miR-21-5p to see if that upregulates inflammatory gene expression. There are two forms of miR-21p that include miR21-5p and miR21-3p. Was miR21-3p also tested? The role of miR23a is unclear as described in introduction. Therefore the description of miR23a should be reduced. The discussion also delves into myriad pathways that miR21 affects that are not being measured in this study. Therefore authors should confined discussion to effects on NF-kB/TLR pathways that have studied in this manuscript.

Ans.: Finally, 5 nM miR-21-5p was used in the transfection experiments. We add this note in the text. Because miR-21-5p was significantly depressed, several inflammatory genes were up-regulated, and prominent cytotoxicity was observed with IHR exposure, we only used miR-21-5p mimic to demonstrate the reversibility. Moreover, miR-21-5p inhibitor may lead to too great cytotoxicity and cause much cell death. The 5p and 3 p forms of miRs can have similar or opposite functions. Only miR-21-5p was tested in vitro and confirmed in the OSA patients in the current study. As suggest, we reduce the description of miR-23a and focus more on the miR-21-TNF-alpha/TLR pathways in the discussion section.

MINOR COMMENTS

Abstract: This should be broken down into Rationale, methods, results and conclusions. If this is not required by the journal, these should be separated into different paragraphs.

Ans.: We will amend it according to the requirement of the journal.

Introduction: The language need to be addressed. few examples are

Para 1, Randomized controlled trials have demonstrated protective effects from... Para 2, and this increase is more pronounced with higher severiies of OSAS indicating that TNFa may be a ... Para 2, line on "In vitro experiments.. " does not make sense

Ans.: As suggest, we address language issues throughout the manuscript.

Reviewer 2 Report

The aim of this paper was to "demonstrate that TLR/TNF-α signaling-related microRNAs may be involved in the pathophysiological cell apoptosis and dysfunction under chronic IHR stimuli in OSA” Among others, the authors found that both miR-21-5P and miR-23-3P expressed differentially in OSA and PS subjects and negatively correlates with apnea hypopnea and oxygen desaturation index. They also found that  TNF-α gene expression increased in OSA patients and positively correlated with apnea hypopnea index. Although the study design is, on the whole, well conducted and the results are clearly reported and well discussed, there are some major concerns regarding the statistical approach used by the authors. 

Usually, fold-change had a positively skewed distribution (mainly due to their non-negative range) and this is quite evident from the scatter plots in figure 1; however, this feature was not properly accounted for by the authors neither in multivariable analysis, where linear models were used, nor in univariate correlations where a Pearson coefficient was computed. In particularly, by looking at figure 1 (d,f,g) it seems that the significance of the correlation was mainly due to the presence of very few (two or three) observations with very high miR-21-5P and miR-23-3P fold change. A non parametric Spearman correlation coefficient should be used to reduce the influence of such observations.  Accordingly, the adjusted analysis should be performed using median regression or probabilistic index models. Moreover, it is not clear why the authors adopted different statistical approaches to analyse, in between groups or within occasions comparisons, the same variables. Figure 3a reports mean (with 95% CI ?) for the miR-21-5P fold changes while in Figure 2d the same variable was summarised using box plot. As these different representations should mimic the different approach used in the statistical comparisons (t-test and Mann-Whitney U test) the author should harmonise the statistical analysis throughout the paper.

Minor remarks. In table 1, male OSA patients should be 33 and not 3. Please, report in table 1 summary statistics for all the variables used in the adjusted analysis. From table 1, it emerged that, for some variables, at least smoking history, there were some missing data. As this factor was used in the multivariable analysis, the authors should report the number of observations available for these models. In Figure 1, please revert panel g and h to make them coherent with the previous panels. Aside from some some specific conditions, Mann-Whitney test is not a test to compare medians. Please, replace in the Statistical Analysis section,  “medians" with “distributions"

Author Response

Response to Reviewer 2

The aim of this paper was to "demonstrate that TLR/TNF-α signaling-related microRNAs may be involved in the pathophysiological cell apoptosis and dysfunction under chronic IHR stimuli in OSA” Among others, the authors found that both miR-21-5P and miR-23-3P expressed differentially in OSA and PS subjects and negatively correlates with apnea hypopnea and oxygen desaturation index. They also found that  TNF-α gene expression increased in OSA patients and positively correlated with apnea hypopnea index. Although the study design is, on the whole, well conducted and the results are clearly reported and well discussed, there are some major concerns regarding the statistical approach used by the authors.

Usually, fold-change had a positively skewed distribution (mainly due to their non-negative range) and this is quite evident from the scatter plots in figure 1; however, this feature was not properly accounted for by the authors neither in multivariable analysis, where linear models were used, nor in univariate correlations where a Pearson coefficient was computed. In particularly, by looking at figure 1 (d,f,g) it seems that the significance of the correlation was mainly due to the presence of very few (two or three) observations with very high miR-21-5P and miR-23-3P fold change. A non-parametric Spearman correlation coefficient should be used to reduce the influence of such observations.  Accordingly, the adjusted analysis should be performed using median regression or probabilistic index models. Moreover, it is not clear why the authors adopted different statistical approaches to analyze, in between groups or within occasions comparisons, the same variables. Figure 3a reports mean (with 95% CI ?) for the miR-21-5P fold changes while in Figure 2d the same variable was summarized using box plot. As these different representations should mimic the different approach used in the statistical comparisons (t-test and Mann-Whitney U test) the author should harmonise the statistical analysis throughout the paper.

Ans.: Thanks for your comments. Actually, Spearman’s correlation was used to correlate two continuous variables. We correct the mistake in making a statistical statement in the method section. Linear regression analysis model is used to obtain adjusted p values, regression coefficients, and 95% CI, because this model is available in most statistical software and the sample size of the case and control groups may be large enough. In addition, median regression would give nearly identical results as linear regression if the assumptions of linear regression are met. As you suggest, we present all the figures in Figure 2 e-n and Figure 3 using box plots to match non-parametric Mann-Whitney U test used in the statistical comparisons.

Minor remarks. In table 1, male OSA patients should be 33 and not 3. Please, report in table 1 summary statistics for all the variables used in the adjusted analysis. From table 1, it emerged that, for some variables, at least smoking history, there were some missing data. As this factor was used in the multivariable analysis, the authors should report the number of observations available for these models. In Figure 1, please revert panel g and h to make them coherent with the previous panels. Aside from some some specific conditions, Mann-Whitney test is not a test to compare medians. Please, replace in the Statistical Analysis section,  “medians" with “distributions"

Ans.: Thanks for your comments. We correct the typo error for the number pf male OSA patients. We correct the missing data in the category variables, and list all statistics of the potential confounding factors used in the multivariate linear regression models. As suggest, we revert panel g and h, and replace “medians” with “distribution” in the method section.

Round 2

Reviewer 1 Report

THe authors have addressed the concerns listed.

Author Response

Thanks for your comment.

Reviewer 2 Report

The authors have responded positively to my previous comments but there still remain some points that need a statistical review.

Maybe due to a misunderstanding on a previous remark of mine (I originally asked to report the number of complete cases that were available for the multivariable analysis), the authors report in table 1 the regression coefficients for all the variables used in the adjusted model. I don’t think these information are of a specific interest for the reader and indeed may complicate the readability of the table. If there are no reasons other than the aforementioned misunderstanding, I suggest to remove the last three columns.

If the Spearman correlation coefficients are reported in Figure 1, please remove the R2 which is simply the square of the Pearson Correlation (and thus is not coherent with the one correctly used by the authors) and, also, do not add the regression line which, again, is based on assumptions that are clearly not supported by the data (normality but, even more, homoscedasticity).

In the Statistical section is now reported that median regression was used to adjust for potential confounding factors. Is this correct? Because, actually, from the authors response, it seems to me that they still prefer to use the linear regression (also in the notes of table 1 they refer to linear regression and not median regression).

Please, carefully check all the results. For instance, the result reported on page 4 (line 107) “regression coefficient 1.913, 95%CI 0.037 to 3.456” cannot be correct, if, as I guess, it refers to the 95% CI of a linear regression coefficient.

Author Response

Response to Reviewer 2 (R2)

The authors have responded positively to my previous comments but there still remain some points that need a statistical review.

Maybe due to a misunderstanding on a previous remark of mine (I originally asked to report the number of complete cases that were available for the multivariable analysis), the authors report in table 1 the regression coefficients for all the variables used in the adjusted model. I don’t think these information are of a specific interest for the reader and indeed may complicate the readability of the table. If there are no reasons other than the aforementioned misunderstanding, I suggest to remove the last three columns.

Ans.: Thank you for your comments. The number of complete cases that were available for multivariable analysis is 20 for PS group and 40 for OSA group. As suggest, we delete the last 3 columns in Table 1.

If the Spearman correlation coefficients are reported in Figure 1, please remove the R2 which is simply the square of the Pearson Correlation (and thus is not coherent with the one correctly used by the authors) and, also, do not add the regression line which, again, is based on assumptions that are clearly not supported by the data (normality but, even more, homoscedasticity).

Ans.: As suggest, we remove the R2 and regression line.

In the Statistical section is now reported that median regression was used to adjust for potential confounding factors. Is this correct? Because, actually, from the authors response, it seems to me that they still prefer to use the linear regression (also in the notes of table 1 they refer to linear regression and not median regression).

Ans.: Thanks. We correct this error.

Please, carefully check all the results. For instance, the result reported on page 4 (line 107) “regression coefficient 1.913, 95%CI 0.037 to 3.456” cannot be correct, if, as I guess, it refers to the 95% CI of a linear regression coefficient.

Ans.: Thanks for your comment. In the comparisons of TNF-alpha, miR-21, and miR-23a gene expression levels between the PS and OSA groups, t-test was used initially. If the p value was <0.05 by t-test, then we used multivariate linear regression analysis model and incorporated all potential confounding factors to get adjusted p values along with regression coefficient and 95%CI. Thus, we reported the adjusted p values, linear regression coefficients, and their 95%CI.